# Understanding incentive preferences of community health workers using discrete choice experiments: a multicountry protocol for Kenya, Uganda, Bangladesh and Haiti

Smisha Agarwal [ORCID],[1] Udochisom Anaba,[2] Timothy Abuya,[3] Richard Kintu,[4] Alain Casseus,[5] Sharif Hossain,[6] Melvin Obadha [ORCID],[7] Charlotte E Warren[2]

For numbered affiliations see end of article.

**Correspondence to**
Dr Smisha Agarwal;
smagarwa@jhsph.edu

## ABSTRACT

**Introduction** There is a renewed global interest in improving community health worker (CHW) programmes. For CHW programmes to be effective, key intervention design factors which contribute to the performance of CHWs need to be identified. The recent WHO guidelines recommends the combination of financial and non-financial incentives to improve CHW performance. However, evidence gaps remain as to what package of incentives will improve their performance in different country contexts. This study aims to evaluate CHW incentive preferences to improve performance and retention which will strengthen CHW programmes and help governments leverage limited resources appropriately.

**Methods and analysis** A discrete choice experiment (DCE) will be conducted with CHWs in Bangladesh, Haiti, Kenya and Uganda with different levels of maturity of CHWs programmes. This will be carried out in two phases. Phase 1 will involve preliminary qualitative research including focus group discussions (FGDs) and key informant interviews to develop the DCE design which will include attributes relevant to the CHW country settings. Phase 2 will involve a DCE survey with CHWs, presenting them with a series of job choices with varying attribute levels. An orthogonal design will be used to generate the choice sets for the surveys. The surveys will be administered in locally-appropriate languages to at least 150 CHWs from each of the cadres in each country. Conditional and mixed multinomial logit (MMNL) models will be used for the estimation of stated preferences.

**Ethics and dissemination** This study has been reviewed and approved by the Population Council's Institutional Review Board in New York, and appropriate ethics review boards in Kenya, Uganda, Bangladesh and Haiti. The results of the study will be disseminated through in-country dissemination workshops, meetings with country-level stakeholders and policy working groups, print media, online blogs and peer-reviewed journals.

## INTRODUCTION

The dearth of health workers, especially in areas where they are most needed, continues to be an ongoing challenge for health systems

### Strengths and limitations of this study

- ► Assessment of multistakeholder perspectives to identify important job attributes will provide policy-relevant and actionable choice options to be tested in the study.
- ► Draws evidence from multiple countries with strong country-level stakeholder involvement including ministries of health, non-governmental organisations and donor agencies.
- ► In countries with multiple cadres of community health workers (CHWs), the study includes formative interviews with at least two cadres of CHWs to understand their perspectives.
- ► As with other discrete choice experiments, a key limitation is that the job alternatives presented to the CHWs are hypothetical.
- ► The results have limited generalisability across the countries given the variation that tends to exist in the needs of CHWs across cadres, terrain and job responsibilities.

globally. The WHO forecasted a global shortage of approximately 18 million health workers by 2030.[1] Recruitment and training of community health workers (CHWs) provides tremendous potential to address this gap in human resources as well as ensure access to basic health services where the formal sector falls short.[2 3] The Alma Ata Declaration (1978), considered a critical twentieth century public health milestone, highlighted the key role of CHWs in advancing 'health for all'.[4] Consequently, CHW programmes became the cornerstone for rapidly expanding primary healthcare (PHC) services for poor populations.[5] Over the 1980s and 1990s, several pilot CHW programmes reported substantial improvements in healthcare outcomes, demonstrating the potential contribution of CHWs.[6 7] However, most national level CHW

programmes were variably implemented and reported several challenges that limited the effectiveness of these programmes.[8–10] Scaled CHW programmes had high levels of attrition and overall reduced quality of care. It was argued that the poor performance of these national level programmes was not due to the failure of the concept of CHWs but due to the lack of ongoing training, supervision, logistical and financial support, as well as weaker linkages to the health system.[11] Often CHWs were unpaid volunteers and were not accountable to the health system. Where strategies to expand care through CHWs were employed as an alternative rather than a complement to professional care, the results were poor.[12] Several debates about selective versus comprehensive PHC approaches ensued.[10] The recent Astana Declaration (2018) has once again emphasised the critical role of PHC in advancing Universal Health Coverage (UHC).[13] Once again, the potential contribution of CHWs to supporting UHC has come to the forefront.

As governments respond to this wave of renewed enthusiasm for CHW programmes, it is critical to be responsive to the lessons from the past. Evidence from studies conducted over the past five decades have identified significant contributions as well as challenges of CHW programmes. Bhutta *et al* conducted a systematic review of evidence to recommend several behavioural and promotional interventions which can be effectively delivered by CHWs with limited training or trained CHWs along the continuum of maternal healthcare. These include the promotion of reproductive health services and family planning, appropriate care seeking and antenatal care during pregnancy, and skilled care for childbirth.[14 15] Additionally, CHWs can play a critical role in several preventive and treatment interventions to support adequate maternal and child nutrition and vitamin supplementation, identification of high-risk pregnancies and childhood illnesses, and early management of preterm labour, malaria in pregnancy, infections and malnutrition in children.[14] However, it is also well-established that CHW programmes can only be effective when CHWs are appropriately recruited, sufficiently trained and adequately supported. Understanding the preferences of CHWs is core to facilitating their retention and improved performance. Broadly, factors affecting the motivation and performance of CHWs can be categorised as: intervention design factors that include but are not limited to incentives, training, supervision, monitoring and evaluation mechanisms[16–18]; and contextual factors like the community context, sociocultural factors, work environment, economic context and health system policy and practice.[15 19 20] While there is some research on the programmatic inputs that contribute to strong CHW programmes, further research on the appropriate combination of these inputs can help governments leverage limited resources appropriately. The 2018 WHO guideline on health policy and system support to optimise CHW programmes identified this as a key research gaps and concluded that 'evidence is not sufficiently granular to allow recommendation of specific forms of interventions, for example … which bundle of financial and non-financial incentives are most effective'.[21]

In many low/middle-income countries (LMICs), CHWs are often funded by government or donors.[22 23] Historically, the commitment to consistently fund comprehensive CHW programmes has wavered; often, in preference for vertical programmes.[19] While in the early years CHWs were thought of as 'volunteers', the importance of integrating these health works into the formal health sector, including the provision of adequate remuneration as well as other incentives has been widely researched and documented.[24–26] The 2018 WHO guideline on CHWs recommended that practising CHWs are remunerated for their work 'with a financial package commensurate with the job demands, complexity, number of hours, training and roles that they undertake'. It further elaborated that incentives can include financial remuneration, including performance-based incentives, as well as non-financial incentives.[21] This was also captured in the Astana Declaration as 'We will create decent work and appropriate compensation for health professionals and other health personnel working at the primary health care level … continue to invest in the education, training, recruitment, development, motivation and retention of the PHC workforce'.[13]

However, barriers to instituting these recommendations in practice remain. First, there is no agreement on strategies that would best support adequate incentivisation of CHWs. Second, with respect to health worker compensation, the health sector human resource policies and practices vary considerably across countries. Third, given the optimal ratio of CHW to population, the question of affordability of scaled CHW programmes to governments remains. A study of CHW programme costs in 37 sub-Saharan African countries estimated that when CHWs are paid an equivalent of US$80/month by existing central government healthcare budgets, the median relative cost of a CHW programme would be 27% of the national healthcare budget.[27]

This study aims to address this critical gap in understanding the combination of financial and non-financial incentives that will support CHWs, improve their work satisfaction and retention, and consequently influence the performance of CHW programmes. The proposed study is being conducted by Frontline Health Project through Population Council, and Johns Hopkins Bloomberg School of Public Health, supported by Bill & Melinda Gates Foundation, in partnership with United States Agency for International Development, UNICEF and Integrating Community Health non-governmental organisation (NGO) partners in Bangladesh, Uganda, Kenya and Haiti. The Frontline Health Project works with global stakeholders to contribute to the objective of advancing evidence for community health. Given the active leadership and participation of donors, NGOs and governments for this study, we expect that the results will aid donors and national policy-makers to develop

training and remuneration policies that are conducive to high performance of CHW programmes, as well as maximise the effective use of limited resources.

The theoretical underpinning of our approach comes from evidence that suggests that the context within which CHW work—including sociocultural factors, support from the health system, relationship with the community, respect and recognition, among others—all contribute to their motivation to work as CHWs, and subsequently their performance.[19 28] For the purpose of this study, we consider these factors, as well as any direct financial compensation as 'incentives'. To elicit the incentive preferences of CHWs, we propose the use of discrete choice experiments (DCEs). DCEs present respondents with two or more hypothetical choices and ask them to select their preferred alternatives.[29] The alternatives are defined by two or more attributes and levels. In this study, we propose to present CHWs with alternatives with different job attributes (eg, salary, promotion, recognition, etc) that have been systematically identified through stakeholder interviews and a review of literature. Based on the selection of preferred alternatives, the influence of each attribute on the choice can be estimated.[30] Furthermore, trade-offs respondents are willing to make can be quantified specifically the marginal willingness to accept (WTA) monetary compensation.

DCEs are based on Lancaster's consumer demand theory and random utility theory (RUT). According to Lancaster's theory, individuals derive benefit or value (utility) from the attributes of a good or service under consideration rather than the good/service itself.[31 32] RUT generally suggests that individuals are rational decision-makers aiming to solve an optimisation problem. They assign perceived utility to each alternative and will therefore choose the one that derives them the highest utility among a set of mutually exclusive alternatives.[33–35] These utilities are contingent on the attributes of the alternative and the characteristics of the decision-maker. However, utilities are latent but researchers can draw conclusions about them from the choices respondents make in a DCE survey.

This approach has been used in comparable studies involving CHWs in India and healthcare workers in LMICs.[36 37] Prior studies have used DCE to model different policy interventions on the recruitment of nurses to rural areas in Kenya, South Africa and Thailand.[38] In Mozambique, DCE was used to elicit the job preferences of non-physician health professionals.[39] In Tanzania and rural Vietnam, DCE has been used to understand how to make jobs in rural areas more attractive to medical students and doctors, respectively.[40 41] In Uganda, DCE was used with volunteer CHWs in family planning programmes in 2011 to examine factors related to their motivation. The study identified recognition in form of t-shirts and badges, a mobile phone and social prestige as some of the core elements associated with CHW motivation.[42 43]

## METHODS AND ANALYSIS

We will use a DCE to examine the job attributes that influence the overall satisfaction and motivation of CHWs in Bangladesh, Haiti, Kenya and Uganda. For the study, we focus on a set of factors that are amenable to national policy and programmatic changes. The study will be conducted in two phases. In the phase 1, we aim to identify a set of attributes that are realistic, pragmatic and meaningful to CHWs and policy-makers, and in the phase 2, we will prioritise critical attributes associated with job satisfaction from the CHWs' point of view. The study design considers the varying context across the four countries, as well as the multiple cadres and roles of the CHWs in each country. The study will be conducted between June 2019 and October 2020.

### Study setting and participants
**Bangladesh** has 8 administrative divisions, 64 districts and 491 subdistricts (upazilas). Each rural area within an upazila is divided into union parishads and mouzas, which are further divided into villages. An urban area in an upazila is divided into wards, and mohallas within each ward.[44] The smallest units within rural areas are the villages and CHWs operate at this level. Bangladesh has several cadres of health workers operating at the community level. Health assistants (HAs), family welfare assistants (FWAs) and community healthcare providers (CHCPs) are commonly referred to as CHWs in the public sector, and they work within the purview of PHC centres known as union health and family welfare centres. Currently, the government is also recruiting multipurpose volunteers and prepaid volunteers in areas where vacancies and workload is higher. In addition, there are CHWs in the non-governmental systems with various names. For example, BRAC's CHWs include the *Shasthya Shebikas*, which has been one of the largest cadres of CHWs in Bangladesh. However, currently the scale and function of the *Shasthya Shebikas* is being revised owing to the high attrition rate of these workers.[45]

Based on discussions with the Directorate General of Family Planning (DGFP) and Directorate General of Health Services (DGHS), the focus of the study will be on the two government-supported CHW cadres: FWAs (supported by the DGFP) and HAs (supported by the DGHS). The study will be conducted in the four divisions of Sylhet, Rajshahi and Khulna, as well as in Chittagong Division's Cox's Bazaar area. The upazilas within each division will be identified through further discussions with the DGFP and DGHS.

**Haiti** is divided into 10 geopolitical regions and subdivided into 42 states (arrondissements).[46] Each arrondissement is further divided into 142 communes in total which comprises 571 communal sections. The health system is overseen at the national level by the Ministry of Public Health and Population (MSPP) which provides policies and standards for service delivery at all levels (including at community level). The District Health Unit, supported by the Health Department Directorate, coordinates,

supervises and monitors care delivery at community level.[47] The primary CHCPs are the 'agents de santé communautaire polyvalent' or ASCP which have been working informally in Haiti for years, but were more formally established by the MSPP in 2015 under one title and clear job description.[47]

This study will take place in three communes located in two arrondissements: one in Artibonite region and the other in Centre region. In Centre region, the study will occur in Mirebalais commune, Mirebalais arrondissement. In Artibonite region, the study will occur in Petite Riviere de L'Artibonite commune and Verettes commune. Study sites and communal sections were selected in collaboration with Zanmi Lasante who have a strong collaboration and presence with the ASCPs in Haiti as well as the MSPP.

**Kenya** is made up of 47 counties with a population projection of more than 40 million. The healthcare system in Kenya is based on the 2010 Kenyan Constitution,[48] which operates a decentralised system of government with the national government and 47 county governments. Kenya has a hierarchical four-tier healthcare system at the community, primary care, secondary referral and tertiary referral levels with community health volunteers (CHVs) operating at the community level. The community services comprise all community-based demand creation activities and health services organised around a comprehensive community strategy defined for the health sector. The primary care services comprise all dispensaries, health centres, and maternity and nursing homes in both public and private sectors. The county referral services include hospitals operating in and managed by a county both public and private which form the county referral system.[49] Every 1000 household (approx. 5000 people) are covered by one community health unit (CHU), comprising a community health committee (CHC) and five community health extension workers (CHEWs). CHEWs supervise CHVs who are embedded within CHUs. CHVs are supposed to be supervised at the community level by CHCs.

In Kenya, the study will take place in Bungoma and Kilifi counties. For phase 1, CHVs, CHEW's county level community focal person and the national stakeholders who works closely with CHV will be interviewed. For phase 2, CHVs will be the primary survey respondent in the two counties. Kilifi is a coastal county with mixed rural urban populations, and Bungoma is an agrarian county in Western Kenya. These counties represent a set of diverse vulnerabilities within which CHWs operate. Within each of these counties, two subcounties will be selected in consultation with the county health management teams.

**In Uganda,** the Health Sector Development Plan highlights the critical shortage of health workers, with a ratio of 1.55 health workers per 1000, compared with the WHO recommendations of 2.28 per 1000.[24] Since 2001, in Uganda, village health teams (VHTs) were established to support community health through health promotion and education activities, mobilisation of communities for utilisation of health services and treatment of illnesses

at the community level. In 2014, based on a comprehensive VHT assessment, it was recommended that the VHT strategy is overhauled, along with consideration of a standardised approach for incentives for all VHTs.[50] In response to these findings, the Ministry of Health (MoH) is in the process of discussing a new CHEW strategy in the recent years. Under this new strategy, CHEWs will be adequately trained in promotive and basic curative health services, and their role will encompass supervising the VHTs. However, at the time of conducting the study, the CHEW policy is pending cabinet approval. In addition to the VHTs, Uganda also have CHWs who are supported by NGOs. Hence, for this study, we focus on two cadres in Uganda: CHWs and VHTs. CHWVHTs, policy-makers and CHW/VHT supervisors will be interviewed during phase 1. Based on detailed discussions with the MoH and other stakeholders, the study will be conducted across eight districts: Lira, Mayuge, Wakiso, Ntungamo, Kabale, Arua, Kabarole and Nakapiripiriti, to facilitate greater representation of the cadres of CHWs/VHTs across the country. These counties were selected based on considerations for the types of terrain and tribes where CHWs serve.

### Phase 1: sampling and approach to inform the development of attributes and levels

In the phase 1, an initial set of policy-relevant, realistic and actionable attributes will be identified based on a review of secondary literature, as well as primary research involving FGDs with a sample of CHWs, their supervisors and national-level stakeholders involved in making decisions about CHW programmes in the country. The attributes and attribute levels will be established separately for each country, based on information collected through the literature review and this formative phase. Discussion topics will include, but not be limited to, expected remuneration, working conditions, availability of supplies and equipment, amount of training, supervision, job role and expected support from the health system and community. Participants will be probed to identify additional priority areas. The FGD will use a nominal group technique where the facilitator will ask the participants to think about aspects of the job that are most important to CHWs as well as actionable by policy-makers. Participants will silently generate a list of attributes, and then be asked to state a single attribute to the group.[51] This will be recorded on a board. This will continue until saturation of attributes is reached. Following this, a discussion to identify the levels of the attributes will be initiated. Participants will be asked to rank order the identified attributes and their levels privately based on their preferences. While there are no restrictions on the number of attributes that can be included in a DCE, in practice, fewer than 10 attributes are selected to reduce cognitive burden on the respondents thereby avoiding inconsistent responses, lexicographic behaviour and attribute non-attendance (where respondents do not consider all attributes in making their decisions).[52] Definitions of each attribute will be defined, appropriate to the setting,

and any overlap or correlation between the attributes will be avoided. Base levels for each attribute will be established to reflect the prevailing working conditions in the country for government-supported CHWs. Additional levels will be then determined to represent a reasonable improvement from the base level. Levels will be chosen to reflect a range of situations that CHWs might realistically expect to experience.

An estimated number of four FGDs with five to seven national and county/subcounty level policy-makers, four to six FGDs with CHW supervisors and approximately six to eight FGDs with CHWs (each cadre) will be conducted in each country. These numbers will vary with the context in each country with the categories of actors involved with community health system as well as the subregional levels where the study will be conducted. Additionally, for policy-makers who cannot be convened in a group setting, the same set of interview questions will be asked to them as key informants in an in-depth interview format. All participants must be aged 18 years and/or with the ability to provide consent. There are no eligibility criteria related to gender or marital status. Policy-makers will be identified by the study team and other relevant partners and CHWs and their supervisors will be recruited through CHUs/PHC facilities in select counties/districts. To ensure that the study takes a ground-up approach, the first set of interviews will be conducted with CHWs and supervisors. The study team will review the results of these interviews to then align the discussion with the policy-makers and other stakeholders to reflect the priorities of the CHWs.

## Prioritising attributes and determining choice sets

Attributes and levels identified in phase 1 across all the interviews will be finalised by the study team based on the discussion transcripts. To ensure that the final set of attributes and levels represents a pragmatic and desirable set, they will be further vetted with external stakeholders such as ministry representatives and NGO leaders working in the CHW space. Efforts will be made to have clear definitions of each attributes and avoid conceptual overlaps across attributes and levels.

A fractional factorial design, which only takes a subset of choice sets, will be generated as too many choice sets place cognitive burden on respondents.[29] An orthogonal design will be generated using Sawtooth Software Inc (Sequim, Washington, USA),[53] where each attribute will be independent of each other and each attribute level will occur equally often (balanced).[35]

Sawtooth Software Inc. To generate the experiment will be unlabelled, full profiles and consist of three alternatives, that is, two job alternatives and an opt-out option.

The DCE questionnaire will include questions about respondent demographics, content and years of training as a CHW, socioeconomic status, and current level and quality of supervision. The order of the choice sets will be randomised to avoid positional bias. In each questionnaire, a dominant alternative will be included where one job option is superior to the other on all characteristics so

that internal consistency and rationality of the responses can be considered during analysis.[29] Finally, the questionnaires will be pretested with a subset of 20–30 respondents and changes made to the content and wording of the questions to account for any conceptual overlap and lack of clarity.

## Phase 2: sampling and approach for the DCE

In each country, the questionnaire will be translated into local language(s). The survey will be administered face-to-face to the CHWs by trained data collectors on tablets using the Open Data Kit software.[54] First, the purpose of the study will be explained. The data collectors will explain the meaning of each attribute to the CHWs to support comprehension of the choice tasks. Each CHW will consider a set of job alternative and will be asked to choose an option they prefer. They will also have to option to opt-out that is, choose neither of the job alternative presented to them.

Sample size calculation in DCE studies mostly rely on rule-of-thumb estimates.[51] For estimation of a DCE model, Lancsar and Louviere[30] suggested that if respondents receive the same design, one rarely requires more than a sample size of 20 respondents. Other studies have shown that sample sizes of 40–100 respondents or up to 300 respondents may be adequate for reliable statistical analysis.[55] The sample size calculation for this study is guided by the rule-of-thumb approach proposed by Johnson and Orme, as well as expected number of choice sets and pragmatic considerations around time and budget to conduct the study across the four countries.[56 57] This approach was selected as it provides a better method of sample size calculation over other techniques such as those proposed by Lancsar and Louviere. According to Johnson and Orme, the minimum sample size $N$ needed to detect differences in the preferences is dependent on the number of analysis cells $c$, number of choice sets $t$ and number of alternatives $a$ as shown in the following equation:

$$N > 500c / (t \times a)$$

Assuming a main effects model, then $c$ will be equal to the largest number of levels for any of the attributes.

If we assume that respondents will face a maximum of 18 choice sets, the largest number of levels of any attribute in a main effects model is four, and that each task has two alternatives (excluding the opt-out), then using the above equation, we derive a minimum sample size of 56. Therefore, we estimate that recruiting a minimum of 150 CHWs per cadre per country to be more than sufficient to detect differences in preferences. A total of approximately 900 respondents across the four countries will be recruited for phase 2 (Bangladesh 300; Haiti 150; Kenya 150; Uganda 300). Based on the total sample size, an equal number of CHWs will be recruited per geographic area in discussion with the county/district health management teams to get broad representation of the types of CHWs.

## Public and patient involvement

Phase 1 of the study is structured to formally gather feedback from the broader community that supports CHWs, including national and international stakeholders and donors, other health officials, CHW supervisors and CHWs themselves. Opinions and experiences of these stakeholders are critical to shaping the questionnaire design and approach for the phase 2 of the study. Health officials at the facilities will be involved in the identification and recruitment of the CHWs in phase 2 of the study. Study results will be disseminated through in-country workshops where health officials involved during the design and recruitment phase will be invited to participate.

## Analysis plan

Conditional logit model will be used as the base model as it is congruous with RUT .[33] This will model the probability of choosing one job alternative over another given the differences in the attribute levels. Preference heterogeneity will be explored by introducing interactions between the attributes. Furthermore, the relative importance CHWs place on the job attributes will be quantified. Additionally, where feasible, marginal WTA estimates will be computed which is the amount of money CHWs are willing to accept to compensate for the worsening or improvement of an attribute.[35] The monetary attribute will be monthly salary while for volunteer CHWs, it will be stipends.

However, one of the shortcomings of the conditional logit model is that the assumption of the independence of irrelevant alternatives (IIAs) must hold. This is as a result of the error terms being assumed to be independently and identically distributed following extreme value type 1 (Gumbel) distribution. To relax IIA and further explore preference heterogeneity, a MMNL model will be used. MMNL presumes that some or all the parameters are randomly distributed with a particular probability distribution.[33] MMNL will result in means which represent the choice probabilities and SD which denote heterogeneity. Stata 16 software will be used in the analysis.[58]

## Limitations of the study in this context

Intrinsic and extrinsic factors influencing CHW incentive preferences are diverse and real-life decisions change over time; therefore, the relative importance of job preferences may vary with time and this needs to be considered when designing CHW programmes and policy interventions. We also recognise that CHW preferences can be different based on their organisational affiliations and prior experiences working as CHWs. While the study aims to capture a sample of CHWs across few areas in each country, the sample is not nationally representative of the preferences of all CHWs. We recognise that a comprehensive job package for CHWs may comprise many more factors that can be adequately captured by this study design. Finally, the results of this study will constitute stated preferences, reflecting responses to hypothetical scenarios. Decision makers will need to consider information from DCEs on CHW as part of the inputs to improved CHW performance and retention while making policies.

## ETHICS AND DISSEMINATION

This study has been reviewed and approved by the Population Council's Institutional Review Board (IRB) in New York, the AMREF-Health Africa Ethics and Scientific Review Committee in Kenya, Zanmi Lasante's IRB (Haiti) and the Higher Degrees Research Ethics Committee at Makerere University College of Health Sciences in Uganda, and the Bangladesh Medical Research Council. Dissemination activities will include sharing of findings through policy briefs, workshops, online blogs in community fora and peer-reviewed journal articles. The research approach is inherently designed around the need for research uptake by policy-makers. Involvement in the research from the beginning is the overarching strategy for promoting research uptake. To ensure there is significant demand for the final study findings, we will debrief policy-makers and other health system stakeholders at pivotal points in the initial stages as well as at the end of the study. We will share non-technical summaries of research results with all the partners for dissemination including the study's key partners in the MoH. If there is enough interest, appropriate dissemination will be conducted with policy-makers in half-day workshops and to facilitate translation of the research findings in national CHW agenda.

**Author affiliations**
[1]Johns Hopkins University Bloomberg School of Public Health, Baltimore, Maryland, USA
[2]Population Council, Washington, District of Columbia, USA
[3]Reproductive Health Program, Population Council, Nairobi, Kenya
[4]Pathfinder International, Kampala, Uganda
[5]Zanmi Lasante, Croix-des-Bouquets, Haiti
[6]Population Council, Dhaka, Bangladesh
[7]Health Economics Research Unit, KEMRI-Wellcome Trust Research Programme, Nairobi, Kenya

**Correction notice** This article has been corrected since it was published. The affiliation for Melvin Obadha has been updated.

**Contributors** SA conceptualised the study and led the writing of the first draft of the protocol. UA assisted with relevant background search and writing. CEW designed the overall front-line health worker study, TA, RK, AC, SH adapted the protocol for Kenya, Uganda, Haiti and Bangladesh, respectively. MO developed the statistical approach for the study. All authors supported the development of the research design and reviewed the final manuscript.

**Funding** This study is funded by the Bill & Melinda Gates Foundation. The funding body has no role in the study design, data collection, analysis or interpretation of data.

**Competing interests** None declared.

**Patient consent for publication** Not required.

**Provenance and peer review** Not commissioned; externally peer reviewed.

**ORCID iDs**
Smisha Agarwal http://orcid.org/0000-0002-7407-0066
Melvin Obadha http://orcid.org/0000-0003-1632-2410

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
