## [Reviewer comments · BMJ Open]

ARTICLE DETAILS

TITLE (PROVISIONAL)	Understanding Incentive Preferences of Community Health Workers using Discrete Choice Experiments: A multi-country protocol for Kenya, Uganda, Bangladesh and Haiti
AUTHORS	Agarwal, Smisha; Anaba, Udochisom; Abuya, Timothy; Kintu, Richard; Casseus, Alain; Hossain, Sharif; Obadha, Melvin; Warren, Charlotte

VERSION 1 – REVIEW

REVIEWER	Aurelie Brunie Scientist, Health Services Research, FHI 360 United States
REVIEW RETURNED	05-Sep-2019

GENERAL COMMENTS	The study design is sound and has many strengths, including the multi-country focus, the exploratory, sequential approach, and the inclusion of the perspectives from multiple stakeholders in the qualitative component informing the design of the DCE. The use of a DCE is appropriate to inform policy-making on this topic because it is difficult or not possible to observe actual choices and understanding trade-offs is important in designing optimal packages of incentives. The background is interesting and comprehensive but more details on the methods would be useful for those who may refer to this protocol in the future. I tried noting some areas where additional information could be beneficial for replication. If there are space limitations, since this is published as a protocol, I would encourage the authors to shorten the background to make space for the methods. As noted by the authors, there is substantial operational and contextual diversity across CHW programs. I believe it would be useful to the reader to streamline the presentation of the four country contexts and to bring greater clarity to some of the key characteristics of these programs, including whether CHWs are currently paid or volunteers, their expected coverage, responsibilities, and literacy level. In Uganda, I was not clear on what CHW referred to, since it seemed to be different from VHTs and CHEWs. It seems that the focus is on a single cadre in each of three countries, and two in Bangladesh. Since one might expect different cadres to operate in different conditions and to possibly have different preferences, I am wondering why the sample size in Bangladesh was not increased accordingly to allow for a differential exploration of FWAs and HAs (alternatively, please clarify if there is a rationale for expecting a certain degree of similarity). On a related note, there seems to be differences in the
--

	geographic scope of the design across countries. While I understand that the primary goal is to introduce some variation rather than obtain a nationally representative sample, it would be good if the authors could offer a justification for the sample size being the same (especially for the qualitative component, due to possibly different saturation points). More justification for the sample size for the qualitative component would generally be helpful. The qualitative component will improve the content validity and reduce the risk of omitting important attributes. It would be helpful to others if the authors could clarify the planned sequencing of the literature review and qualitative phase. Specifically, I would be interested to know if the study team is planning some standardization of the list of attributes covered in the qualitative interviews within each country. The approach seems to be structured around listing relevant attributes, which makes sense. I am wondering if there are plans to probe for additional attributes that may have been identified in the literature review but may not have been mentioned spontaneously to ascertain their importance and enhance the ability to collate results across qualitative data collection events in the same country. The background notes several possible, related outcomes that can be affected by bundles of incentives, including motivation, satisfaction, performance and retention. It would be helpful to include more details on how this is being operationalized as part of the two components. For the qualitative component, it is mentioned that moderators will ask about aspects of the job that are "important and actionable." For the DCE, the probe is to choose a job that would "best support them given their circumstances." Please clarify how this relates to the outcomes and overall objective of the study, and what will be done to ensure that this is not interpreted differently by different participants within the same country. I am also wondering if the study team is envisioning sub-group analyses to examine possible differences in stated preferences across CHWs within the same country based on current performance or number of years of service. I much appreciated the discussion around design aspects of the DCE, including the number of attributes, appropriate framing, alignment of base levels with prevailing conditions and avoiding overlap or correlation, as well as the emphasis on selecting reasonable and actionable improvements. We conducted DCEs in three countries, including one with CHWs in Uganda in 2011 and two with potential family planning users in Nigeria and India. Based on our experience with these populations (relative to DCEs conducted with more educated health workers), we found that including more than 5-8 attributes or 10-12 choice tasks could prove challenging and that including pictures alongside the description of attributes and levels was valuable. Because DCE results are contingent on the range of levels included for each attribute, we also learned that, while being realistic, it could be important to encourage stakeholders to consider a sufficient range of possible levels. To this point, I am wondering what is being planned to reconcile possibly different inputs from CHWs and other stakeholders in defining the range of attributes and levels and balancing being realistic with not being overly conservative.
--	--

	While I am familiar with the DCE methodology, the protocol may benefit from additional review by a statistician prior to publication. Could the authors please include their specific assumptions in applying the rule-of-thumb approach by Johnson and Orme for the DCE sample size (number of tasks, alternatives and levels or products of levels)? The authors mention that fatigue may set in after 18 choice tasks, but as noted above, a more conservative number might need to be considered. How would marginal willingness to accept estimates be derived where CHWs are volunteers (or is this when it would be considered not to be feasible)? I agree with the authors' observations that there will be differences in what is realistic or appropriate across countries and was not entirely clear on how a regional or multi-country analysis may then be feasible (although it is mentioned that it will only be conducted if appropriate). I would expect that, even if this is a multi-country protocol, the primary implications would be for decision-making within each country rather than in terms of generalizable knowledge across countries. The authors raised the questions of affordability and sustainability of incentive packages in the background, which are very salient. I am wondering if there are any plans to cost possible intervention packages based on DCE results as an additional step. Lastly, I agree with the comment that DCE results will be valuable as part of a larger package of results and inputs to inform policy since their predictive validity is still largely unknown. I wish the team all the best with study implementation and look forward to seeing the results!
--	---

REVIEWER	Maryse Kok KIT Royal Tropical Institute, The Netherlands
REVIEW RETURNED	11-Sep-2019

GENERAL COMMENTS	This is a well-written research protocol on a relevant and important study, focusing on incentive preferences of CHWs in different contexts. I recommend the protocol to be published. The authors could consider to clarify the following:  - With regard to Uganda, which cadres of CHWs will be included in phase 2 of the study? VHTs and CHWs, and for the latter, which types? - To add a short elaboration about how motivation, satisfaction, motivation and performance are believed to be linked, from a theoretical point of view. - To add Brunie et al. 2014 and 2016 as references, who also conducted a DCE with voluntary CHWs in Uganda.
--

VERSION 1 – AUTHOR RESPONSE

Reviewer: 1

The study design is sound and has many strengths, including the multi-country focus, the exploratory, sequential approach, and the inclusion of the perspectives from multiple stakeholders in the qualitative component informing the design of the DCE. The use of a DCE is appropriate to inform policy-making on this topic because it is difficult or not possible to observe actual choices and understanding trade-offs is important in designing optimal packages of incentives. The background is interesting and

comprehensive but more details on the methods would be useful for those who may refer to this protocol in the future. I tried noting some areas where additional information could be beneficial for replication. If there are space limitations, since this is published as a protocol, I would encourage the authors to shorten the background to make space for the methods.

1. As noted by the authors, there is substantial operational and contextual diversity across CHW programs. I believe it would be useful to the reader to streamline the presentation of the four country contexts and to bring greater clarity to some of the key characteristics of these programs, including whether CHWs are currently paid or volunteers, their expected coverage, responsibilities, and literacy level. In Uganda, I was not clear on what CHW referred to, since it seemed to be different from VHTs and CHEWs. It seems that the focus is on a single cadre in each of three countries, and two in Bangladesh. Since one might expect different cadres to operate in different conditions and to possibly have different preferences, I am wondering why the sample size in Bangladesh was not increased accordingly to allow for a differential exploration of FWAs and HAs (alternatively, please clarify if there is a rationale for expecting a certain degree of similarity).

Thank you for noting these discrepancies in the manuscript. We have clarified that VHTs and CHW (health workers supported by NGOs) will be interviewed in Phase 2 in Uganda. The sample size is 150 per cadre per country. Since we include 2 types of CHW cadres in Uganda and Bangladesh, 300 CHWs will be interviewed in Phase 2 in these countries. This has been clarified in text. As for the details on key characteristics of these programs, in most cases, guidelines around structure, compensation and responsibilities are not present or are dated, or if they are available, the actual implementation (which has implications for our approach) has a lot of variations. The protocol cannot cover this level of detail for each country; however, we do intent to elaborate on these details in the results manuscripts which will have scope for deeper country-level analyses.

2. On a related note, there seems to be differences in the geographic scope of the design across countries. While I understand that the primary goal is to introduce some variation rather than obtain a nationally representative sample, it would be good if the authors could offer a justification for the sample size being the same (especially for the qualitative component, due to possibly different saturation points). More justification for the sample size for the qualitative component would generally be helpful.

The proposed sample size is an estimate of the category of study participants in each country. Therefore, we agree that the number and type will vary in each study setting. we have amended the section that describes the sampling for qualitative study in page 14 as follows:

“An estimated number of about 4 FGDs with 5-7 national and county/subcounty level policy makers, 4-6 FGDs with CHW supervisors and ~6-8 FGDs with CHWs (each cadre) will be conducted in each country. These numbers will vary with the context in each country with the categories of actors involved with community health system as well as the sub regional levels where the study will be conducted. Additionally, for policymakers who cannot be convened in a group setting, the same set of interview questions will be asked to them as key informants in an in-depth interview format.”

3. The qualitative component will improve the content validity and reduce the risk of omitting important attributes. It would be helpful to others if the authors could clarify the planned sequencing of the literature review and qualitative phase. Specifically, I would be interested to know if the study team is planning some standardization of the list of attributes covered in the qualitative interviews within each country. The approach seems to be structured around listing relevant attributes, which makes sense. I am wondering if there are plans to probe for additional attributes that may have been identified in the literature review but may not have been mentioned spontaneously to ascertain their importance and enhance the ability to collate results across qualitative data collection events in the same country.

Thank you for this comment. We do plan to generate more ideas from attributes even beyond what might be available in literature, as well as use the attributes identified in the literature for probing responses. In page 13 we have provided details and clarified the process.

4. The background notes several possible, related outcomes that can be affected by bundles of incentives, including motivation, satisfaction, performance and retention. It would be helpful to include more details on how this is being operationalized as part of the two components. For the qualitative component, it is mentioned that moderators will ask about aspects of the job that are “important and actionable.” For the DCE, the probe is to choose a job that would “best support them given their circumstances.” Please clarify how this relates to the outcomes and overall objective of the study, and what will be done to ensure that this is not interpreted differently by different participants within the same country. I am also wondering if the study team is envisioning sub-group analyses to examine possible differences in stated preferences across CHWs within the same country based on current performance or number of years of service

The overall objective of the study is to understand incentive preferences of CHWs with the aim to affect their motivation and retention. To understand their preference, we seek to ask about job aspects that are “important”; to understand whether these job aspects can be realistically implemented from a policy stand-point, we ask them to reflect on the “actionability” of these attributes. As with all areas of qualitative and quantitative enquiry, there is always the possibility that different respondents understand the questions differently. To address this, we use the following research procedures- train the data collection team on the intent of the questions, include examples in the questionnaire to clarify the questions, pilot test the survey tools, monitor the results to identify discrepancies and finally include this possibility in our reporting.

Yes, interactions will be included to explore preference heterogeneity. This has been captured under the analysis plan section by including the following statement *“Preference heterogeneity will be explored by introducing interactions between the attributes”*.

5. While I am familiar with the DCE methodology, the protocol may benefit from additional review by a statistician prior to publication. Could the authors please include their specific assumptions in applying the rule-of-thumb approach by Johnson and Orme for the DCE sample size (number of tasks, alternatives and levels or products of levels)? The authors mention that fatigue may set in after 18 choice tasks, but as noted above, a more conservative number might need to be considered. How would marginal willingness to accept estimates be derived where CHWs are volunteers (or is this when it would be considered not to be feasible)? I agree with the authors’ observations that there will be differences in what is realistic or appropriate across countries and was not entirely clear on how a regional or multicountry analysis may then be feasible (although it is mentioned that it will only be conducted if appropriate). I would expect that, even if this is a multi-country protocol, the primary implications would be for decision-making within each country rather than in terms of generalizable knowledge across countries.

We agree that sample size calculations in DCEs are challenging when you don’t have the final number of attributes and levels. Therefore, Johnson and Orme’s approach was preferred as it provides a better method of sample size calculation over other techniques such as those proposed by Lancsar and Louviere. We have updated the sampling and approach of DCE section as follows *“According to Johnson and Orme, the minimum sample size N needed to detect differences in the preferences is dependent on the number of analysis cells c , number of choice sets t , and number of alternatives a as shown in the following equation;*

$$N > 500c / (t \times a)$$

Assuming a main effects model, then c will be equal to the largest number of levels for any of the attributes. If we assume that respondents will face a maximum of 18 choice sets, the largest number of levels of any attribute in a main effects model is four, and that each task has two alternatives (excluding the opt-out), then using equation 1, we derive a minimum sample size of 56. Therefore, we estimate that recruiting a minimum of 150 CHW per cadre per country to be more than sufficient to detect differences in preferences”

We also agree that 18 choice tasks might still place significant cognitive burden on the respondents. We intend to have fewer than 18 choice sets depending on the final number of

attributes and levels we will be able to derive. 18 is just an upper limit. Therefore, we paraphrased the sentence in the prioritizing attributes and determining choice sets section to read as follows *“A fractional factorial design, which only takes a subset of choice sets, will be generated as too many choice sets place cognitive burden on respondents.”*

Moreover, where a monetary attribute will be present, then marginal willingness to accept estimates will be computed. For CHWs, the monetary attribute will be stipends. Employed CHWs are paid salaries. Computation of marginal WTA estimates will only be feasible in the presence of a monetary attribute. We have stated this in the data analysis plan section as follows *“Additionally, where feasible, marginal WTA estimates will be computed which is the amount of money CHWs are willing to accept to compensate for the worsening or improvement of an attribute (Cite). The monetary attribute will be monthly salary while for volunteer CHWs, it will be stipends.”*

We do agree that there will be heterogeneity across countries making a multi-country comparison a challenge. Therefore, the main aim is not generalizability across settings rather the primary implications would be for the policy makers within each country as you have stated. Therefore, we have deleted the sentence that states that a comparison of results across the different countries will be conducted.

6. I much appreciated the discussion around design aspects of the DCE, including the number of attributes, appropriate framing, alignment of base levels with prevailing conditions and avoiding overlap or correlation, as well as the emphasis on selecting reasonable and actionable improvements. We conducted DCEs in three countries, including one with CHWs in Uganda in 2011 and two with potential family planning users in Nigeria and India. Based on our experience with these populations (relative to DCEs conducted with more educated health workers), we found that including more than 5-8 attributes or 10-12 choice tasks could prove challenging and that including pictures alongside the description of attributes and levels was valuable. Because DCE results are contingent on the range of levels included for each attribute, we also learned that, while being realistic, it could be important to encourage stakeholders to consider a sufficient range of possible levels. To this point, I am wondering what is being planned to reconcile possibly different inputs from CHWs and other stakeholders in defining the range of attributes and levels and balancing being realistic with not being overly conservative.

We appreciate this insight from your prior experiences. Are there published works we could refer to for this? We anticipate having a similar challenge reconciling the Phase 1 results from interviews with CHWs and policymakers. However, we do think that these perspectives are both critical and need to be accounted for. Given that CHW incentives cannot be restructured without a commitment of financial resources, our plan is to present the results from the CHWs to the stakeholders and engage in a discussion to arrive on a final list of attributes. This is presented in the manuscript as *“To ensure that the study takes a ground-up approach, the first set of interviews will be conducted with CHWs and supervisors. The study team will review the results of these interviews to then align the discussion with the policymakers and other stakeholders to reflect the priorities of the CHWs.”*

Published guidance suggests that 18 choice sets are the maximum before fatigue sets in; however, we plan on closer to 12 choice sets. We will consider inclusion of pictures or other memory aids based on the results of pilot testing the tools.

7. The authors raised the questions of affordability and sustainability of incentive packages in the background, which are very salient. I am wondering if there are any plans to cost possible intervention packages based on DCE results as an additional step. Lastly, I agree with the comment that DCE results will be valuable as part of a larger package of results and inputs to inform policy since their predictive validity is still largely unknown. There have been discussions around costing the possible intervention packages based on the DCE, but no financial commitments to support that have been made yet, and we do not propose undertaking costing as part of this protocol.

Reviewer: 2

This is a well-written research protocol on a relevant and important study, focusing on incentive preferences of CHWs in different contexts.

I recommend the protocol to be published. The authors could consider to clarify the following:

1. With regard to Uganda, which cadres of CHWs will be included in phase 2 of the study? VHTs and CHWs, and for the latter, which types?

Thank you for catching this discrepancy. VHTs and CHWs will be included in Phase 2. We have clarified that CHWs in this case refer to NGO-supported CHWs.

2. To add a short elaboration about how motivation, satisfaction, motivation and performance are believed to be linked, from a theoretical point of view.

We have elaborated briefly on this linkage and added references.

3. To add Brunie et al. 2014 and 2016 as references, who also conducted a DCE with voluntary CHWs in Uganda.

Thank you for directing us to these studies. These have been referenced in the protocol now.